# Lessons to Learn from the Gut Microbiota: A Focus on Amyotrophic Lateral Sclerosis

**DOI:** 10.3390/genes13050865

**Published:** 2022-05-12

**Authors:** Ana Cristina Calvo, Inés Valledor-Martín, Laura Moreno-Martínez, Janne Markus Toivonen, Rosario Osta

**Affiliations:** 1Department of Anatomy, Agroalimentary Institute of Aragon (IA2), Embryology and Animal Genetics, University of Zaragoza, Calle Miguel Servet 13, 50013 Zaragoza, Spain; inesvalledormartin@gmail.com; 2Centro de Investigación Biomédica en Red de Enfermedades Neurodegenerativas (CIBERNED), Aragon Institute for Health Research (IIS Aragon), Calle Miguel Servet 13, 50013 Zaragoza, Spain; lauramm@unizar.es (L.M.-M.); osta@unizar.es (R.O.)

**Keywords:** amyotrophic lateral sclerosis, gut dysbiosis, gut microbiota, postbiotics, prebiotics, probiotics, transgenic SOD1G93A mice

## Abstract

The gut microbiota is able to modulate the development and homeostasis of the central nervous system (CNS) through the immune, circulatory, and neuronal systems. In turn, the CNS influences the gut microbiota through stress responses and at the level of the endocrine system. This bidirectional communication forms the “gut microbiota–brain axis” and has been postulated to play a role in the etiopathology of several neurodegenerative diseases, including amyotrophic lateral sclerosis (ALS). Numerous studies in animal models of ALS and in patients have highlighted the close communication between the immune system and the gut microbiota and, therefore, it is possible that alterations in the gut microbiota may have a direct impact on neuronal function and survival in ALS patients. Consequently, if the gut dysbiosis does indeed play a role in ALS-related neurodegeneration, nutritional immunomodulatory interventions based on probiotics, prebiotics, and/or postbiotics could emerge as innovative therapeutic strategies. This review aimed to shed light on the impact of the gut microbiota in ALS disease and on the use of potential nutritional interventions based on different types of biotics to ameliorate ALS symptoms.

## 1. Introduction

Sequencing of the human genome has greatly improved our understanding of our genetic variability and improved our knowledge on the etiopathogenic origin of diseases. Despite this, causes of many diseases of a multifactorial nature such as amyotrophic lateral sclerosis (ALS) remain largely unknown, which has contributed to the fact that there is still no effective treatment that slows down the degenerative process in ALS patients.

ALS, one of the most-known rare diseases, is an adult-onset, devastating, neurodegenerative disease characterized by the loss of cortical, brain stem and spinal motor neurons and by progressive muscle atrophy. The majority of ALS patients, close to 90%, are classified as sporadic cases (sALS) of unknown cause, while the rest of the patients are identified as familial cases (fALS), since they carry a mutation in one of the genes related to the disease. These genes include TAR DNA binding protein (TARDBP/TDP-43), superoxide dismutase 1 (SOD1), FUS RNA binding protein (FUS/TLS), or the most recently discovered gene C9orf72-SMCR8 complex subunit (C9orf72) with ALS-causing hexanucleotide repeat expansions [1,2,3]. Most fALS cases have an autosomal dominant inheritance pattern, except for some autosomal recessive patterns that have been reported in the case of specific mutations in the ALS gene, especially detected in families from Germany, Asia, and North Africa [4].

The onset age of fALS follows a normal Gaussian distribution and takes place a decade earlier than the onset of sALS, which is approximately 60 years of age. In rare cases, disease onset may occur before the age of 25 years, and this juvenile form of the disease is almost always genetically determined. The crude annual incidence rate of ALS in the general European population has risen during the last five years and it is estimated to be two to three cases per 100,000 person-years [5,6]. The incidence is higher among men than among women in a ratio as high as 2.6:1, although some recent studies have reported a more balanced gender ratio [6]. Spinal-onset ALS is more common among men when compared to women, particularly by the age of 70–80 years. Albeit, though the disease occurrence decreases rapidly after 80 years of age, the median survival ranges between 3 and 5 years from symptom onset [5,6,7]. The multisystemic nature of ALS has prevented the exact identification of its first steps of neurodegenerative progress. ALS results from a complex array of factors, including oxidative stress, mitochondrial dysfunction, endoplasmic reticulum stress, dysregulated transcription and RNA processing, dysregulated endosomal trafficking, impaired axonal transport, protein aggregation, excitotoxicity, apoptosis, inflammation, and genetic susceptibility. To date, increased phosphorylated neurofilament heavy chain in cerebrospinal fluid and neurofilament light chain in plasma, serum, or cerebrospinal fluid can support the diagnosis and prognosis evaluation in ALS patients. However, increased levels of these neurofilaments are also observed in other neurodegenerative diseases. Consequently, the combination of neurofilaments with other methods rather than their use alone in the diagnosis and prognosis of ALS could be valuable. Actually, the diagnosis of ALS follows the revised El Escorial, the Airlie-House criteria, and the Awaji-shima criteria, to discriminate any pathologic evidence of other disease processes that might explain upper or lower motor neuron degeneration [8,9]. More recently, an emerging Gold Coast criteria has proven to be more sensitive and specific for characterizing progressive muscular atrophy and for excluding primary lateral sclerosis as a form of ALS. This criteria also investigates findings that exclude alternative diseases and, therefore, it can enable a more accurate diagnosis, ruling out confusions from the early stages of the disease [10]. Another useful instrument for the evaluation of the functional status of ALS patients is the revised ALS Functional Rating Scale (ALSFRS-r). This parameter can be used to monitor the functional state of the patient over time, and it is strongly related to survival and ALS prognosis. In addition, recent neuroimaging tools and novel algorithms have paved the way to the conversion of ALSFRS-r into clinical stages to predict disease progression [11,12]. A great proportion of the molecular mechanisms that are potentially involved in ALS pathogenesis have been discovered through mouse models, especially the ones expressing human SOD1 with ALS-associated mutations, such as SOD1G93A [13]. In particular, the transgenic SOD1G93A mice show a phenotype that resembles the disease progression in patients and therefore, they are one of the most used in preclinical trials.

Despite the above-mentioned clinical findings, an early diagnosis and prognosis of the disease is urgently needed. In the field of neurodegenerative diseases, the composition of the bacterial microbiota is a novel and growing area of research interest, as communication between the brain and gut microbiota is essential to maintain homeostasis. In this “gut microbiota–brain axis”, the central nervous system (CNS), the neuroendocrine and neuroimmune systems, autonomic nervous system, enteric nervous system, and intestinal microbiota signal bidirectionally to each other [14]. Signals from the brain can influence the motor, sensory, and secretory activities of the gastrointestinal tract and, conversely, visceral messages from the gastrointestinal tract can influence brain function [15]. One group of the main molecular messengers that support the communication between the nervous system and the gastrointestinal are the short-chain fatty acids (SCFA), which are metabolic products of anaerobic bacteria by fermentation of dietary fiber [16]. SCFA are water soluble and can be easily absorbed or transported into cells. In particular, SCFA can modify the recruitment of circulating leukocytes to the inflammatory site, preventing inflammation and immune responses through the modulation of regulatory T lymphocytes (Treg). This bidirectional communication has been postulated to have a role in the etiopathology of various neurodegenerative diseases including ALS [16]. Albeit, although earlier immune responses need further investigation, numerous studies indicate that inflammation contributes to neurodegenerative progress in ALS. In fact, alterations in peripheral monocytes, neutrophils, and various T-cell populations correlate with the rate of ALS progression. Such modifications have been identified even in presymptomatic ALS mutation carriers [17] suggesting an active contribution of immune alterations to the pathology and degenerative progress of the disease.

Considering the above evidence, this review aimed to shed light on the potential implication of the gut microbiota in ALS, aiming to recapitulate our current understanding on the crosstalk between gut microbiota and disease progression. In addition, a wide range of potentially therapeutic nutritional interventions are reviewed in order to summarize to which extent the modulation of gut symbiosis can revert or ameliorate a pathological state. Finally, a comparative analysis between the gut microbiota in humans and mice was explored to better understand to which extent translational research on microbiota can be performed in translating these important disease models to human applications.

## 2. Materials and Methods

A systematic literature search was conducted on PubMed and Web of Science (WoS). All related articles published up to February 2022 were considered for inclusion. Search queries were as follows: Search 1: Amyotrophic Lateral Sclerosis [mh] AND (Gut Microbiota [mh] OR Intestinal microbiota [mh] OR Gut Microbiome [mh] OR Intestinal Microbiome [mh] OR Gut Flora OR Intestinal Flora [mh] OR Gastrointestinal Microbiota [mh] OR Gastrointestinal Flora [mh]. Search 2: Amyotrophic Lateral Sclerosis [mh] AND (Dietary Supplements [mh] OR Probiotics [mh] OR Synbiotics [mh] OR Prebiotics [mh] OR Postbiotics OR Fecal Microbiota Transplantation [mh] OR Fecal Transplant [mh]). Moreover, other relevant references of articles were also reviewed for general introduction purposes. The search results for ALS and terms synonymous with “Gut microbiota” amounted to 33 entries in PubMed and 19 in WoS. Results for ALS and “dietary supplements” amounted to 50 entries in PubMed and 276 in WoS.

## 3. Is the Gut Microbiota in Humans and Mice Really So Different?

Mammals are colonized by an enormous number of microorganisms, composed mainly of non-pathogenic bacteria, which we know as microbiota. Microbiota associates with different tissues and organ systems, notably the gastrointestinal tract, which constitutes the greatest human microbial barrier. The number of genes expressed from the human microbiota is estimated to surpass 150 times that of the human genome [18,19,20] and microbiota-derived proteins and metabolites strongly contribute to the proper immune, endocrine, and metabolic function. Consequently, it is not surprising that alterations in the gut microbiota are related to various pathological states such as obesity and diabetes, as well as autoimmune, inflammatory, and neurological diseases. The bacteria (archaea and eukarya) that have symbiotically colonized the human gastrointestinal tract during the evolution of the human species are known as “gut microbiota” [21]. This symbiotic community modulates the development and homeostasis of the CNS through the immune, circulatory, and neural systems. In turn, the CNS influences the gut microbiota through stress responses and at the level of the endocrine system. Additionally, the gut microbiota plays an important role in the maintenance of gut integrity and functions as a source of energy that can protect the host from the pathogen invasion [22]. Consequently, an imbalance or disturbance of the bacterial composition in the gut, intestinal dysbiosis, may lead to an alteration of these protective mechanisms.

During the last two decades, it has been feasible to analyze, in depth, the composition of the gut microbiota by using high-throughput sequencing methodology. In this sense, sequencing of the bacterial 16S ribosomal RNA (rRNA) gene provides a useful tool to distinguish among different species since it is present in all bacteria and archaea. Albeit, although the analysis of shorter regions in 16S rRNA that contains nine hypervariable regions can enable a more detailed study, errors can also be present in a higher proportion than using whole-genome shotgun metagenomics [23]. To date, databases based on the understanding of the bacterial populations that constitute the human microbiome have gathered relevant information about the microorganisms that inhabit the human gut. Such databases include MicrobiomeBD [24], the NIH Human Microbiome Project [25], and the MDB: Microbiome Database [26].

Differences in the microbiota between and within species are largely derived from different dietary habits. One of the benefits of using murine models is that the diet can be experimentally controlled, although it could also be argued that experimentation on a highly defined diet may not well reflect natural conditions. However, a recent study suggests that the gut microbiota in mice and humans shares similar features at the genus level and, therefore, could show significant similarities from the functional point of view [27]. Indeed, some clinical studies are in accordance with the previous findings obtained in mice [28,29]. In light with this, there is still a strong challenge to translate the findings obtained in animal models to patients, considering the complexity and the difficult management of human diseases [30], especially neuroinflammatory diseases such as multiple sclerosis, Alzheimer disease, Parkinson’s disease, and autism spectrum disorders that share altered molecular targets with ALS. In these cases, the etiopathogenic nature of the disease promotes a leaky intestinal permeability, leading to a sustained inflammation that could, in turn, affect the blood–brain barrier integrity. Therefore, nutritional guidelines need to be fitted to each inflammatory disease, which would affect the gut microbiota balance in a different way.

From a descriptive point of view, sequencing analysis in healthy C57BL/6J mice identified 37 core bacterial genera that could help to differentiate age, sex, and state (health or disease) and, consequently, could be used to determine the host phenotype. Among these, 17 genera were also found in the human gut microbiota, which indicates the degree of similarities between the two species. In addition, it is remarkable that the 37 core bacteria could be classified into five subgroups with different functions. In the first and largest subgroup, butyrate-producing and anti-tumor immunity-promoting bacteria were identified, suggesting that this subgroup could be involved in the modulation and balance of the immune system. The second subgroup were composed of carbohydrate-utilizing and lactate- and/or acetate-producing bacteria that could play a relevant role to counteract inflammation. The third subgroup also included butyrate-producing bacteria and it could favor the stability of the bacterial population in case of a dysbiosis. The fourth subgroup was comprised of amino acid-utilizing bacteria, while the last subgroup gathered bacteria that were able to metabolize cellulose into acetate and hydrogen [27] (Table 1).

There are five main core genera of bacteria in the mice gut with different functionalities, although these groups are correlated with each other [27].

It was proposed that this complex network of functionally precise subgroups of bacteria can help in the maintenance of a balanced gut microbiota and, at the same time, can function as therapeutic targets to ameliorate or revert pathological processes by modulating the bacterial population in question in the gut. As an example, gut inflammation could be ameliorated in different mouse models, including the transgenic SOD1G93A mouse for the study of ALS [31,32], by increasing the numbers of butyrate-producing bacteria from subgroup one. Interestingly, administration of bacteria from subgroups one and two in infant and early adulthood mice could promote a more mature and healthy core gut microbiota [33].

Regarding the human gut microbiota, more than two thousand species have been identified, 386 of which reside in mucosal regions, such as the oral cavity and the gastrointestinal tract [22]. The gut microbiota in humans shares similarities in composition and function with other bacterial populations in different body sites, which is in accordance with the functional redundancy also observed in mouse gut microbiota [21,26]. In contrast to mice that are generally housed with controlled conditions, the bacterial composition in the human gut can be greatly influenced by environmental factors, which suggest that human microbial signatures may vary considerably in different regions. Despite this fact, the functional redundancy may balance the differences in bacterial composition in the gut and, thus, may allow common therapeutic strategies to reshape the gut microbial community that can produce similar metabolite profiles.

During the first years of life, the human gut microbiota has a low diversity and is mainly composed of *Actinobacteria* and *Proteobacteria*. However, during the first years of life, the composition and diversity of the gut microbiota approximates that found in the adults [34] where the gut microbiota mainly produces SCFA, propionate, butyrate, and acetate, these being the most common molecules found in a proportion of 1:1:3, respectively, in the gastrointestinal tract. *Bacteriodetes* (*Akkermansia muciniphila*) mostly generate propionate, while *Actinobacteria* and *Firmicutes* (*Eubacterium rectale*) are the main producers of butyrate. SCFA can modulate the production of cytokines to preserve the epithelial integrity, but they also present specific functionality once they are released from the gut. Propionate promotes gluconeogenesis in the liver and exerts a beneficial effect on the modulation of β-cells, while butyrate and acetate can activate lipogenesis. Butyrate, on the other hand, is an energy source for colonocytes and can promote anti-inflammatory and anti-cancer activities [22], which is in accordance with the functional role of bacterial population of the two first subgroups in the mouse gut microbiota [27]. In fact, both human and mouse share the main dominant bacterial genera *Prevotella*, *Bacteroides*, *Clostridium*, *Eubacterium*, *Parabacteroides*, *Ruminococcus*, *Faecalibacterium*, *Roseburia*, *Blautia*, and *Coprococcus*, which are also present in pigs and in cynomolgus macaque [35].

## 4. Implication of Gut Dysbiosis in Animal Model of ALS

ALS models such as SOD1G93A are important tools to investigate presymptomatic alterations as well as the onset and progression of the disease in genetically homogenous populations and controlled environment. SOD1G93A mice show increased intestinal epithelial permeability due to elevated levels of proinflammatory cytokines, which directly affect the junctions of the intestinal epithelium [36]. This “leaky gut” phenomenon facilitates the translocation of bacterial toxins through the intestinal lumen to the bloodstream from where they may further translocate to different tissues and organs, including the CNS. This may be especially relevant in ALS, because the blood–brain barrier in patients and in animal models for the disease is negatively affected [37]. The dysfunction of the intestinal barrier can promote the passage of toxins from the intestinal lumen to the blood and increase the immune response in addition to the increase of circulating lipopolysaccharides (LPS). This could have a pivotal role in the pathogenesis of ALS, since LPS favor inflammatory processes that affect the function of the afferent neurons of the vagus nerve [32,38]. All of these factors that are linked to ALS can be favorably modulated by some microbial populations and their metabolites in the intestine. In line with this, an abnormal density of Paneth cells in the intestinal crypt and increasing levels of IL-17 inflammatory cytokines have been also reported in transgenic SOD1G93A mice, suggesting that altered gut microbiota could affect disease progression in this murine model [39].

Another potential factor that could influence the pathogenesis of ALS is the dysregulation of bacterial populations in the gut. The observed imbalance between two main phylae of bacteria, *Firmicutes* and *Bacteriodetes*, could contribute to the alteration of pro-inflammatory factors by changes in the levels of endotoxins and exotoxins secreted by these bacteria, and hence, affect the progress of the disease [40]. Interestingly, one published work in a SOD1G93A animal model [41] proposes that the population of *Firmicutes* bacteria is diminished before the onset of symptoms and possibly reflects a disease-stage-dependent (presymptomatic vs. symptomatic) modulation of microbiota in this model. In line with this, early changes in the composition of the gut microbiota in at least ten phylae of bacteria in SOD1G93A mice have also been described and been correlated with a more aggressive progression of the disease in mice, which can be exacerbated under germ-free conditions or after antibiotics administration [40,42]. Nevertheless, *Akkermansia muciniphila* and nicotinamide supplementation improve motor function and survival in the mice [40,42]. Furthermore, dietary butyrate increases the abundance of at least some members of *Firmicutes* population in the SOD1G93A model [32]. Collectively, this suggests that the *Firmicutes*/*Bacteriodetes* ratio may be dynamically and environmentally altered during the disease’s progression, and that the SOD1G93A related abnormalities in the gut microbiota may be corrected using relatively simple dietary interventions. In this sense, it could be possible to consider that targeting the gut microbiota could counteract the inflammatory response and metabolic alterations inherent in ALS.

The imbalance of the bacterial populations in the gut of SOD1G93A mice is also reinforced in a detailed longitudinal study performed in this model that highlighted an early alteration before the disease onset [43]. *Actinobacteria*, *Bacteriodetes*, *Firmicutes*, and *Verrucomicrobia* communities were found to be dysregulated during early stages of disease progression, not only in fecal pellets but also in colon and ileum sections, exacerbating the metabolic disturbances. In parallel, leukocyte infiltration into the CNS, myeloid activation, and increased global 5-Hydroxymethylcytosine levels in the brain of the SOD1G93A mice were observed along disease progression, suggesting that epigenetic changes may possibly be considered as a new factor affected by the gut-brain axis [43].

Although most studies have been carried out in SOD1G93A mice, new models, such as those based on mutations in C9orf72 gene have been under much attention recently. Related to the microbiota, a reduction of the immune-stimulating bacteria in C9orf72-mutant mice as well as the transplantation of gut microflora in this murine model protected it from systemic inflammation and autoimmunity mediated by the gut microbiota, reinforcing the connection between this microbiota and ALS at preclinical level [44].

## 5. Role of the Gut Microbiota in ALS Patients

Studies carried out to characterize potential alterations the gut microbiota in ALS patients are often contrasting due to the heterogeneity observed among the different cohorts of patients studied, to the different experimental methodologies, and, last but not least, due to the protocol of conservation of fecal samples. The first two studies conducted on gut microbiota in ALS patients showed a different expression and a lower diversity in the genus and class level with respect to healthy controls [45,46]. The ratio between *Firmicutes* and *Bacteriodetes*, the two major phylae contributing to the gut microbiota, has been proposed to alter in different pathological states [47]. In the first study conducted by Fang and coworkers [45], a decreased *Firmicutes*-to-*Bacteriodetes* ratio and a decrease in beneficial bacteria, such as *Anaerostipes*, *Lachnospiraceae* and *Oscillobacter*, were observed in ALS patients and not in healthy individuals, suggesting that this altered gut microbiota could influence disease progression. In the other study, an alteration of the *Firmicutes*-to-*Bacteriodetes* ratio was observed in ALS patients, accompanied with increased fecal inflammatory markers [46], suggesting that the imbalance of the gut microbiota could be associated with the pathogenesis of the disease. These findings were also in accordance with a more recent study performed in patients with probable or definite ALS and their caregivers, in which *Firmicutes* at the phylum level and *Megamonas* at the genus level, as well as gene profile related to different metabolic pathways, were decreased with respect to healthy individuals [47]. However, another study using 16S rRNA analysis reported a higher abundance of *Firmicutes* and a lower abundance of *Negativicutes* and *Bacilli* in patients with respect to control individuals, albeit no significant differences were found in the levels of metabolites, such as human endotoxin, SCFA and NO_2_-N/NO_3_-N, and c-aminobutyric acid [48].

However, the above studies were based on very few subjects in each group. More recently, a study performed in an Italian cohort of patients using the selective quantification of bacterial species and yeast by quantitative PCR also indicated an imbalance of the gut microbiota in ALS patients, which showed increasing *Escherichia coli* and *Enterobacteria* populations [49]. In contrast, a study conducted in a German cohort of patients found no significant alterations in the gut microbiota of ALS patients with respect to healthy controls, except for a differential abundance of *Ruminococcaceae* at the genus level [50]. In this study, pyrosequencing of the 16S rRNA gene and modeling of predicted metagenomes were used but the findings obtained in this study were consistent with those obtained in another study, performed in an American cohort of patients, in which metagenomic sequencing was used. In this latter study, no association between individual taxa and clinical variables was found, albeit a reduced abundance of butyrate-producing bacteria, such as *Eubacterium rectale* and *Roseburia intestinalis* was observed in ALS patients [51], which was also in accordance with a study focusing on three CNS disorders, including ALS [52]. Studies on gut microbiota in other neurodegenerative diseases such as multiple sclerosis, Parkinson’s disease, stroke, and Alzheimer’s disease [53,54,55,56] have also provided variable results in patients, probably due to the high inter-individual variability, the difficult management of these diseases, as previously mentioned, and the need of standardizing the methodology and including homogeneous patient groups to avoid bias. It is also important to understand that most of the above-mentioned studies were based on quantitative PCR and pyrosequencing, two methodologies that have been now largely replaced by direct sequencing and platforms with lower error rates [57].

A very recent two-sample Mendelian randomization study based on 98 bacterial genera of the human gut supported the close relationship between the gut microbiota and ALS. In particular, OTU10032, unclassified *Enterobacteriaceae* species-level OTU, and unclassified *Acidaminococcaceae* correlated with a higher risk of ALS. In addition, an increasing relative abundance of OTU4607_*Sutterella* and *Lactobacillales*_ORDER was also found to be related to a higher ALS susceptibility, which could enhance a higher degree of permeability in the gut. These genera were found to be related to γ-glutamyl-related metabolite levels, suggesting that a trans-synaptic, glutaminergic, excitotoxic mechanism could be involved in the pathogenic basis for ALS [58].

Furthermore, a recent translational study that aimed to analyze the human gut profile considering previous findings in transgenic SOD1G93A mice suggested a functional decrease in bacterial genes related to nicotinamide (NAM) metabolism the in stool samples of ALS patients with respect to healthy individuals, which was in clear connection with the results obtained in the murine model [42]. Interestingly, significantly reduced NAM levels were also found in the CSF samples of patients, highlighting that the metabolites produced under gut dysbiosis could finally reach the CNS [42].

Finally, environmental neurotoxins, such as β-Methylamino-l-alanine (BMAA) that has been related to people from Guam Isle, have been proposed as key factors that might be involved in the pathogenesis of ALS disease. Increasing levels of BMAA were observed in the brain tissues of ALS patients of Guam, suspecting that they may have ingested this compound through their diet. Due to the permeability of the gut in ALS patients, BMAA was thought to exert its detrimental effect in the patients. In fact, *Cyanobacteria* and the *Archaea* can produce BMAA, which could promote a higher degree of gut dysbiosis [39].

Altogether these findings in animal models of ALS and ALS patients underline the close crosstalk between gut microbiota and ALS. At the same time, the possibility of modulating the gut microbiota can yield novel therapeutic strategies, particularly interesting in ALS.

## 6. Prebiotics, Probiotics, and Postbiotics as Nutritional Strategies to Balance the Bacterial Population in the Gut

To date, there are five U.S. Food and Drug Administration-approved drugs for ALS, Rilutek, Radicava (Edaravone), Tylotic, and Exservan, which aim to slow the progression of the disease. Rilutek was the first drug to be approved in 1995, to prolong the life expectancy around three months. Tiglutik is the thickened liquid form of Rilutek, while Exservan is the oral film formulation of Rilutek, which has also been approved in Canada, Australia, and Europe, while the United Kingdom approved Tiglutik under the name Teglutik. However, Exservan and Radicava are still under review in many countries [59].

However, increasing dietary therapies using compounds or live organisms that modify inflammatory response, mainly including probiotics and prebiotics, are being lately investigated as an alternative to the “single target” immunomodulatory therapies, as epidemiological evidence suggests a strong correlation among diet and the onset of many neurodegenerative diseases, including ALS [60]. Considering the range of possible diet-based interventions, it is interesting to describe the main groups of live organisms and compounds that may have a modulatory effect on the gut microbiota in murine models and ALS patients (Table 2).

Regarding ALS disease, different probiotics, prebiotics, and postbiotics have been tested in transgenic SOD1G93A mice and ALS patients.

### 6.1. Probiotics

According to the Food and Agriculture Organization of the United Nations (FAO) and the WHO Working Group, live organisms that exert a beneficial effect on the host when administered in adequate amounts are considered probiotics [61]. This beneficial effect is required for either a strain-specific or group level for a probiotic. These organisms can modulate the immune system, metabolism, hypertension, cellular cycle, neurodegenerative and oxidative processes, as well as balance the gut dysbiosis, and they can also generate beneficial metabolites such as vitamins, acid, and SCFA [62]. The most well-known bacterial genera considered as probiotics are *Bifidobacterium* (adolescentis, animalis, bifidum, breve and longum) and *Lactobacillus* (acidophilus, casei, fermentum, gasseri, johnsonii, paracasei, plantarum, rhamnosus, and salivarius). More lately discovered bacterial species such as *Akkermansia muciniphila* and *Faecalibacterium prausnitzii*, together with other butyrate producing bacteria such as *Roseburia* spp. and *Eubacterium hallii*, have been tested in different animal models to ameliorate inflammation in the gut, modulate the immune system, and reinforce the intestinal barrier functionality. In contrast, dead microbes and microbial products and components are not considered probiotics [58]. Interestingly, the positive or beneficial effect of probiotics depends on the strain, dose, and components included in the probiotic product [63].

A detailed study published in *Nature* [40] demonstrated that the intestinal supplementation of *Akkermansia muciniphila*, which is a gut microbe with an important role in intestinal mucin degradation, ameliorated the symptoms of ALS in transgenic SOD1G93A mice. At the level of metabolites, the beneficial effect of gut-supplemented *Akkermansia muciniphila* was demonstrated to rely on increased nicotinamide levels in the CNS of SOD1G93A mice, and it was also demonstrated that systemic nicotinamide levels are downregulated in ALS patients. However, mucin degradation produces SCFA and some SCFA-producing bacteria are negatively affected in ALS models [33,36] and patients [45].

Furthermore, in a prospective longitudinal study performed in ALS patients, the daily administration of a probiotic formulation composed of a mixture of five lactic acid bacteria (*Streptococcus thermophilus*, *Lactobacillus fermentum*, *L. delbrueckii subsp. delbrueckii*, *L. plantarum*, and *L. salivarius*) modulated the bacterial diversity in ALS patients but did not result in changes that would mimic healthy individuals. In addition, this probiotic treatment did not influence on the disease progression monitored by ALSFRS-R scale [64].

### 6.2. Prebiotics

The FAO/WHO Working Group defined prebiotics as a non-digested food component that can exert a beneficial effect in the host by modulating the gut microbiota. The beneficial effect is not always easily predicted but such non-digested components are selectively fermented by commensals and probiotic bacteria in the gut to generate other compounds, such as SCFA or butyrate [65]. Prebiotics can be used as an alternative to probiotics or to support them [63]. Fruit, vegetables, cereals, and many other edible plants are natural sources of prebiotics, such as polyphenols, inulin, pectin, or oligofructose. However, artificial prebiotics are also used, such as lactulose, galactooligosaccharides, fructooligosaccharides, maltooligosaccharides, cyclodextrins, and lactosaccharose [63].

One of the first studies that reported the beneficial effect of the most frequently used prebiotics in transgenic SOD1G93A mice was the study published by Song and coworkers in 2013. Here, the administration of galactooligosaccharides in this animal model delayed the onset of the disease, prolonged the life span in the mice, significantly reduced the motor neuron loss and muscle atrophy, and ameliorated the inflammatory response in the CNS of the SOD1G93A mice [66].

Other prebiotic compounds that are widely used are the polyunsaturated acids. In particular, in a longitudinal study that included five prospective cohorts of ALS patients in the United States suggested that omega-3 polyunsaturated acid intake could delay the onset of the disease [67]. Nevertheless, the dietary supplementation of eicosapentaenoic acid in transgenic SOD1G93A mice at the pre-symptomatic stage accelerated disease progression and shortened the life span in the mice, suggesting that a toxic aldehydic oxidation product of this polyunsaturated acid had increased in the spinal cord of the animals, increasing the reactive microglia [68].

**Table 2 genes-13-00865-t002:** List of probiotics, prebiotics, and postbiotics tested in transgenic SOD1G93A mice and ALS patients.

Dietary Intervention	Microorganism or Compound	Mechanism of Action	Reference
PROBIOTICS	*Akkermansia muciniphila*	Slow disease progression in transgenic SOD1G93A mice and increase nicotinamide levels in CNS	[40,41]
	*Streptococcus thermophilus*, *Lactobacillus fermentum*, *Lactobacillus delbrueckii subsp.**Delbrueckii*, *Lactobacillus plantarum*, *and Lactobacillus salivarius*	Modulation of the bacterial diversity in ALS patients but this diversity was not found similar to the one observed in healthy individuals	[64]
PREBIOTICS	galactooligosaccharides	Delay in the onset of the disease, longer life span in the mice, lower motor neuron loss and muscle atrophy and amelioration of the inflammatory response in the CNS of transgenic SOD1G93A mice, absorption and synthesis of B vitamin in the colon	[66]
	omega-3 polyunsaturated acid (eicosapentaenoic acid)	Short disease progression, increasing microglia in the spinal cord of transgenic SOD1G93A mice	[68]
	omega-3 polyunsaturated acid	Delay in the disease onset in ALS patients	[67]
	curcumin	Reduction of oxidative stress in ALS patients, no change in disease progression	[69]
POSTBIOTICS	butyrate	Rise in Treg lymphocytes, decreased levels of pro inflammatory cytokines, slow disease progression, decreased in the permeability of the gut, balancing of the gut dysbiosis in transgenic SOD1G93A mice	[66]

Among polyphenols, curcumin, which is a non-flavonoid polyphenol, has been also tested in ALS patients to evaluate its neuroprotective and antioxidant properties. Oral supplementation of curcumin in ALS patients decreased the oxidative damage markers, although disease progression was not altered during the 6 month follow-up period [69]. The combinations of probiotics and prebiotics that can act in a synergistic manner are known as synbiotics. The synbiotics exert a synergistic and beneficial effect in the survival and colonization of life-beneficial microorganisms in the gastrointestinal tract in the host and they can modulate not only the bacterial composition in the gut but also the production of microbial metabolites. To date, no information about any study performed in murine models of ALS and/or ALS patients that uses synbiotics to ameliorate disease progression has been reported.

### 6.3. Postbiotics

Postbiotics are the latest members of the biotic family that comprise bioactive compounds produced by food-grade microorganisms in a fermentation process, such as SCFA, microbial fractions, functional proteins, secreted polysaccharides, extracellular polysaccharides (EPS), cell lysates, teichoic acid, peptidoglycan-derived muropeptides, and pili-type structures [70]. Two main types of postbiotics are paraprobiotics and fermented infant formulas that can exert a beneficial effect on the host when administered in adequate amounts.

Importantly, the administration of butyrate increases the levels of Treg lymphocytes in the blood, favoring a decrease in the levels of the inflammatory cytokine IL-17, and slowing down the progress of the disease in SOD1G93A transgenic mice [32].

## 7. Conclusions and Future Directions

The analysis of the role of the gut microbiota in ALS research has yielded relevant findings that can help to better understand key factors related to the bacterial composition and diversity in the gut that could influence in the disease progression and in impaired molecular pathways in animal models and patients. In addition, the possibility of using probiotics, prebiotics, and postbiotics to ameliorate the symptoms and the progression of the disease opens a new field of nutritional interventions that could be easily performed and monitored in future clinical trials. Notwithstanding, future studies are needed to decipher in depth the potential role of the gut microbiota in this complex and multifactorial disease. In this sense, a main challenge needs to be explored to analyze at which extent ALS disease can impact the gut microbiota. The sub-optimal nutritional conditions during the disease progression could influence the gut microbiota, and therefore confounding effects inherent in the disease need to be considered before suggesting key factors that are really involved in the gut–brain axis that and influence ALS disease.

## Figures and Tables

**Table 1 genes-13-00865-t001:** Main core gut genera in mice.

Core Gut Genera		Function
	*Anaerostipes*, *Anaerotruncus*,	
	*Oscillibacter*, *Clostridium XlVb*, *Bacteroides*, *Barnesiella*, *Alistipes*,	
C1	*Helicobacter*, *Saccharibacteria_genera_incertae_sedis*, *Prevotella*,	modulation and balance of the immune system
	*Lachnoanaerobaculum*, *Intestinimonas*, *Roseburia*, *Alloprevotella*,	
	*Rikenella*, *Allobaculum*, *Lachnospiracea_incertae_sedis*,	
	*Pseudoflavonifractor*, *Marvinbryantia*, *Mucispirillum*,	
	*Odoribacter and Acetatifactor*	
C2	*Bifidobacterium*, *Olsenella*, *Lactobacillus*,	amelioration of inflammation
	*Enterorhabdus*, *Parasutterella*, *and Turicibacter*	
C3	*Parabacteroides*, *Flavonifractor*,	the stability of the bacterial population in case of a dysbiosis
	*Clostridium XIVa*, *Blautia*, *and Anaerofilum*	
C4	*Erysipelotrichaceae_incertae_sedis*	amino acid production
	*Eggerthella*, *and Gordonibacter*	
C5	*Ruminococcus*	acetate and hydrogen production

## Data Availability

Not applicable.

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
