# Peer review of "Lessons to Learn from the Gut Microbiota: A Focus on Amyotrophic Lateral Sclerosis"

_genes, 2022, doi:10.3390/genes13050865_

Round 1

Reviewer 1 Report

This is an interesting review about ALS and the potential role if microbiote that could be a therapeutic target for this devastative disorder.

As a whole, few points to underline, but the parts of the manuscript focused on ALS itself need some changes/editing as there are some errors or misleading sentences.

lines 38 to 45: too much genes are listed, some of them probably not really the cause of ALS. It would be more pragmatic, and scientifically correct, to list the 4 major genes implicated in familial ALS, they make consensus: C9ORF72, SOD1, FUS and TARDBP.

line 47: I disagree with the sentence. No ALS is not an age-dependent disorder, only Alzheimer's disease is age-dependent. As written later in the manuscript, there is a drop of incidence after 80s...The sentence should thus be more precise on line 47.

lines 51-52: juvenile ALS is an ALS with an onset before 18 years old, not 30 !! Please correct. If ALS before 30 years represents almost 5% of the cases, as written, juvenile cases are exceptionnal, less than 1% of all ALS cases, and almost always genetically determined-ALS.

line 60: the assertion of the authors "Due to the fact that no effective
biomarkers have been described to date, the prognosis of the disease remains poor" is absolutely false for two reasons. First, many works have shown that NfL is a good and reliable marker of ALS diagnosis and prognosis. Secondly, there is no relationships between the fact that there are no biomarkers and the poor prognosis. I do not really understand the association between those two assertions.

lines 348-353: treatments for ALS. The authors only consider marketed drugs in the USA, and thus is a problem, all the more as they wrote that "Radicava treatment has demonstrated to slow the decline of physical function in ALS patients", and this is false. Only USA decided to market Radicava, while EMEA, considering that only a short and small phase II study was done, no demonstration of effect at all has been done. This is the reason why phase III studies are ongoing.

line 353: the authors are mixing Nuedexta, a drug with only a symptomatic influence, with drugs that could modulate handicap and survival. This is rather misleading and we propose that this drug be removed from the list or cited clearly aside.

line 472: "needed" instead of "need" I guess

Reviewer 2 Report

With this review, the authors want to introduce the concept that there may be a link between the gut microbiota and amyotrophic lateral sclerosis (ALS), as already proposed for neuroinflammatory diseases such as multiple sclerosis (MS), Alzheimer's disease (AD), Parkinson's disease (PD), and autism spectrum disorder (ASD). As with the latter diseases, the authors propose the utility of nutritional interventions for the purpose of balancing the gut microbial population, improving symptoms, and slowing the progression of ALS. Their nutritional strategies are focused on the use of prebiotics, probiotics and postbiotics. 

OUR COMMENTS: 

1)    In our opinion, to suggest that the composition and "functional capacity" (what would that be?) of the gut microbiota could represent a "reliable molecular biomarker" of ALS (lines 78-82), is a rather unlikely and general assertion since an altered gut microbiota can be correlated with several chronic inflammatory diseases. 

2)    Paragraph 3 on the significant similarities between the gut microbiota in mice and humans could be reduced to no more than half or eliminated. The use of animal models is now generally accepted, but they should not be equated with human disease, much less when it comes to the composition of the gut microbiota in mice and humans.  In fact, in the same paragraph (title: “Is the gut microbiota in humans and mice really so didifferent?”), the authors state that "in contrast to mice, which are housed under controlled conditions, the human gut microbiota can vary considerably", without mentioning that one of the first causes of diversity is precisely the different diet of mice and humans. 

3)    Instead of the current paragraph 3, it would be better to consider both dietary habits and lifestyle, inflammation and its origin, as well as the importance of barrier integrity, meaning both the intestinal barrier and the blood-brain barrier. In fact, it has already been reported that as a result of gut dysbiosis there is intestinal inflammation and breakdown of the intestinal barrier. If the intestinal barrier becomes leaky, it allows the passage of microbial and immune cells, endotoxins and cytokines, as well as of incompletely digested food. All these cells and molecules  may cooperate in breaking the blood-brain barrier (BBB) and result in the setup of neuroinflammatory diseases.  In this context, several studies have shown the importance of dietary advises in diseases as  MS, AD, PD e ASD. The authors should take a stand and consider whether the origin of the neuroinflammatory diseases above, as well as the suggested nutritional guidelines, might be the same.

4)  lines 315-319: It is not clear what the purpose of this sentence is.  The assertion that studies of other neurodegenerative diseases have produced mixed results is questionable. It may be that it is not about standardizing methodology or including homogeneous groups of patients, but about taking into account that the composition of the gut microbiota is highly individual. This could also explain the heterogeneity observed in different cohorts of ALS patients studied (lines 278-280).

SOME MINOR COMMENTS IN DETAIL: 

•    Gut microorganisms are not "millions" (line 119) but many more.

•    Enlarge the characters in Tables 1 and 2,

•    Insert some reference when the bidirectional communication called "gut microbiota- brain axis" is mentioned.

•    It would be better to avoid the definition of "commensals" (line 410) or clarify what is meant by it.

•    lines 322-342:  Avoid describing the work cited, and instead to establish what the implications of these studies are. 
